# Characteristics of Adolescent Life Goals in Contemporary China: A Mixed-Methods Study

**DOI:** 10.3390/bs13040326

**Published:** 2023-04-11

**Authors:** Xiaofeng Wang, Rui Fu, Aruna Wu, Dan Li

**Affiliations:** 1Department of Psychology, Shanghai Normal University, Shanghai 200030, China; 2Counseling and Psychological Services Center, Shanghai University of Political Science and Law, Shanghai 201701, China; 3Center for Violence Prevention at Children’s Hospital of Philadelphia, Philadelphia, PA 19146, USA

**Keywords:** mixed-methods, Chinese adolescents, life goals, social change

## Abstract

Adolescence is a developmental period when individuals actively evaluate and construct their life goals. During the past several decades, China has transformed dramatically toward a highly competitive, market-oriented society. Despite a growing interest in exploring the implications of cultural values for youth adjustment in contemporary China, little is known about what life goals are prevalent among Chinese adolescents. This mixed-methods study aimed to identify the key themes of life goals and to examine gender, grade, and urban-rural differences in the identified themes among Chinese adolescents, using quantitative and qualitative methods. Semi-structured interviews were conducted with a sample of 163 middle- and high-school students in urban and rural China. Thirteen key life goal themes were identified; among them, the most mentioned themes were Family Well-being, Academic Excellence, and Personal Happiness. Quantitative results showed grade and urban-rural differences in the adolescents’ endorsement of the themes of life goals. Specifically, more middle schoolers and rural students endorsed life goals that emphasize social belonging and group well-being, whereas more high schoolers and urban students endorsed life goals that underscore individual independence and uniqueness. These results indicated the implications of social change for adolescents’ life goals in contemporary China.

## 1. Introduction

Life goals are the objectives that individuals consciously form to direct their lives over time [1], which represent an inherent desire of human beings to achieve life meaning and purpose [2]. Life goals can be categorized into intrinsic and extrinsic types, and while intrinsic life goals serve to satisfy the basic psychological needs for competence, autonomy, and relatedness (e.g., personal growth, community contributions), extrinsic life goals (e.g., wealth, fame) aim to obtain external rewards and depend on others’ evaluations [3]. In pursuit of intrinsic and extrinsic life goals, individuals ultimately seek to have self-transcendence and a coherent, meaningful life [4].

Previous research on life goals has suggested the relevance of intrinsic and extrinsic life goals across many cultures [3,5]. This universal relevance of life goals echoes the work of Schwartz’s basic human values that serve two main interests: personal uniqueness and interests, as well as social affiliation and the well-being of groups [6]. Also, the universal distinction between intrinsic and extrinsic life goals is aligned with Rokeach’s conceptualization of values comprising terminal and instrumental values [7]. Terminal values refer to desirable and end-state existence and reflect individuals’ life goals, whereas instrumental values are preferable modes of behavior that are means of achieving terminal values and life goals [7]. Taken together, universally recognized life goals stem from self-and group-oriented interests that guide people’s attitudes, emotions, and behaviors.

Adolescence represents a critical period when children’s life goals are being shaped as one major developmental task for identity formation [3]. During this period, individuals begin to consciously identify with compatible values and life goals while maintaining a balance between conflicting goals [8]. However, less research work has focused on life goals during adolescence as compared to other developmental periods, despite the long-term implications of life goals for individuals’ physical and psychological well-being [9,10].

In the literature, there has been some empirical evidence suggesting gender and age differences in adolescents’ life goals. For example, female youth had more intrinsic life goals and a higher sense of personal meaning responsible for one’s life than male students [3,11]. Among Chinese youth, females tend to score higher on caring for others and other group-oriented life goals [4,12]. Regarding age differences, research has shown that older adolescents tend to identify more strongly with self-oriented life goals [13]. This argument has been consistently endorsed in studies among Chinese adolescents. For example, one study showed that Chinese high schoolers identified more with self-oriented life goals and values, such as personal happiness and self-improvement, than middle school students who, in contrast, identified more with group-oriented values, including social equality and family well-being [14,15].

In addition, recent studies have suggested intra-individual changes in personal values and life goals, particularly when individuals are in a changing environment [16]. Exploring such contextual changes is important in societies such as China, given the dramatic societal changes (e.g., urbanization and economic development, geographic mobility, increase in internet users, nuclear family structures, and the expansion of women in the workplace) that have shaped the individuals’ life goals. This has been supported by the increasing quantitative research that indicates the influence of social changes on adolescents’ self- and group-oriented values in contemporary China [17,18]. For example, in the longitudinal study of Chen et al. [12], although girls were more group-oriented than boys at the initial time point, there were no gender differences in the growth patterns in their self and group orientations, suggesting distinctive gender similarities and differences in youth values and life goals attributed to the rise of gender equality and rapid urbanization in contemporary China. However, less is known, from a qualitative perspective, about the specific life goals that Chinese adolescents endorse. The current study sought to address these literature gaps and to provide a more nuanced understanding of the developmental and contextual changes in life goals in Chinese adolescents.

### 1.1. Life Goals in Chinese Adolescents

Self and group orientations represent two fundamental domains of beliefs that individuals hold toward the world in relation to themselves [19]. This categorization is contextualized by differences in beliefs between individualistic (United States, Western Europe) and collectivistic societies (Mexico, China) [20]. Similar to the categorization of basic values, self-oriented life goals are mainly characterized by the independence and uniqueness that motivate individuals to actively make their own choices and decisions [21,22]. In contrast, group-oriented life goals tap into a sense of social affiliation in the group and emphasize the attainment of group benefits and interpersonal relationship harmony [17,23]. In collectivist societies, group interests are generally prioritized over individual interests, and as such, individuals are expected to sacrifice themselves when the two interests are in conflict.

Although developing behaviors that are aligned with self- and group-oriented life goals are regarded as a major task of children and adolescents, they appear to have different implications for development in different societies. Relative to individualistic societies, collectivistic societies tend to emphasize more group-oriented values and less self-orientated values, and children and adolescents are thus expected to prioritize life goals that promote group well-being over those that enhance individual interests [24]. In traditional Chinese society, with Confucianism serving as a predominant ideology for social activities, the expression of personal desires or striving for autonomous behaviors is considered unacceptable, especially when it threatens group harmony [25]. There is supporting literature that group-oriented goals are encouraged in school education and family socialization, particularly in many rural areas in China [18]. In these areas, youth who behave in accordance with these goals are likely to be accepted and approved by peers and adults, which leads to the acquisition of social status, positive social relationships, and psychological adjustment [26].

Despite a growing literature on the influence of self- and group-oriented values and life goals on Chinese adolescents’ development [18,27], it remains largely understudied what specific themes of life goals are entailed within the domains of self and group orientations, respectively. The majority of the current research on the two orientation domains in Chinese adolescents utilized a quantitative approach by which the youth were asked to respond to a scale [27] (e.g., the Children’s Cultural Values Scale) that tapped the individuals’ general cognitive and behavioral preferences for self, versus for others (e.g., “I like to express my own opinions”, “It is important to me to respect decisions made by the group”). This approach, however, precludes investigations of the variations in the youth’s accounts for their life goals that reveal their meaning-making of life purpose in various areas, including school, work, family, leisure, and faith. Capturing the breadth of life goals among youth is crucial because it serves to organize one’s daily goals and behaviors, specifically time, energy, and other resources one directs in different areas of life. This argument is aligned with the literature on youth development of life purpose [28] that calls for the importance of exploring their goals and meaning in life in and beyond the immediate context (e.g., goals related to school and/or eventual career) and of measuring life goals and values by quantitative (e.g., survey instruments) and qualitative methods (e.g., interviews, focus groups).

### 1.2. Life Goals in Chinese Adolescents in the Context of Social Change

According to socioecological perspectives [29], life goals are largely guided by culture, which is subject to continuous historical change. This argument has been elaborated in a growing literature on the important implications of social change for parental attitudes and practices and for their associations with child adjustment, particularly in East Asian societies, such as China and South Korea [30]. Yet, the characteristics of adolescents’ life goals in changing circumstances remain largely unknown, which may serve as a critical mechanism through which social change impacts adolescents’ development.

In a traditional Chinese society with Confucianism serving as a predominant ideological guideline for social activities, the major virtues of Ren (humanity), Yi (righteousness), Li (ritual propriety), and Xiao (filial piety) are highly valued. Among the four virtues, filial piety serves as the root of all Confucian ethics [31]. In the belief of filial piety, children and adolescents are expected to respect their parent’s authority, comply with their expectations and demands, as well as perform well academically [32]. In particular, academic excellence is considered the key to enhancing the status and reputation of the family and to strengthening children’s connectedness with their parents [31].

However, due to its massive economic and social change in recent decades, China has transformed dramatically toward a highly competitive, market-oriented society, particularly in urban regions. Along with the change, new behavioral characteristics, such as social assertiveness and self-confidence, are increasingly accepted and appreciated by individuals, particularly in the younger generation [24]. Indeed, Chen et al. [27] (p. 243) have recently found that urban youth in China endorsed life goals of individual uniqueness and the formation of a “unitary and stable” self that is separate from the social context (e.g., “I enjoy being unique and different from others in many respects”, “I like to behave in my way”). Moreover, emerging research has revealed an increase in the importance that Chinese adolescents ascribe to life goals of pursuing individual pleasure and sensuous gratification [33]. As indicated in Greenfield’s (2009) social change and human development theory [34], globalization and related sociodemographic changes have urged the changes in life goals that facilitated the development of relational interdependence and the attainment of group well-being to those that facilitate the development of individual autonomy, uniqueness, and personal growth [18]. In many developing countries, urbanization as a manifestation of the globalization process has shaped individual cultural beliefs and values to be more self-oriented [23]. Yet, it is also noted that this promotion of self-oriented values does not necessarily lead to a reduction in reduced need for social connectedness and affiliation [35]. Indeed, children and adolescents, in the context of social change, may engage in the constructive process that allows them to integrate different values, such as an integration of self-oriented values into traditional group-oriented values in contemporary China [17,36]. Nevertheless, this integration of different values in the context of social change does not necessarily lead to self- and group-oriented life goals being endorsed to the same extent in urban and rural regions of China [18,35]. As the massive economic and social changes have been largely limited to the urban areas, people in the rural areas have been less exposed to the shift of values and life goals toward self-orientation than those in the urban areas [37]. Therefore, group-oriented goals are viewed as more important and are still highly encouraged in school education and family socialization, which lead to these goals being influential or effective in guiding rural Chinese children’s social interactions and activities [24,26]. On the other hand, self-oriented goals are less appreciated than group-oriented goals in rural areas, and rural youth are less encouraged to develop new behavioral qualities that promote autonomy and self-assertiveness [27]. This argument is supported by Liu et al. [18], that overall, self-orientation was associated with social and emotional adjustments more strongly in urban children, whereas group orientation was associated with adjustments more markedly in rural children. Given the different functional meanings of self and group orientations in urban and rural Chinese children, it will be important to delve into the urban-rural differences in Chinese adolescents’ endorsement of specific themes of life goals identified within self-and group orientations, respectively.

### 1.3. Overview of the Present Study

The main goal of this mixed-methods study was to explore the key themes of life goals in Chinese adolescents and the urban-rural, grade, and gender differences in the youth’s endorsement of life goals. Prior research has indicated the implications of social change for adolescents’ self and group orientations and for the contributions of the two orientations to adolescents’ adjustment in contemporary China [17,18]. However, few studies have examined specific life goals in different areas of life in Chinese adolescents, and even fewer have compared regional differences in their endorsement of life goals, considering that urban children are more exposed to the impact of social changes. Given this gap in the literature, we sought to examine the key themes of life goals in Chinese adolescents and how the prevalence of the endorsed life goals might differ between urban and rural regions. This line of research would help us better understand the contributions of social change in contemporary China in shaping adolescents’ life goals.

This exploratory investigation utilized qualitative semi-structured interviews, an appropriate means by which to collect in-depth, individualized data that are particularly significant to each participant, rather than those that are guided by predetermined categories or measures [38]. This method is particularly useful for broadly exploring themes or patterns within the data that relate to the context-embedded meanings of values [39]. This is conducive to extending our understanding of how ongoing social changes in China are involved in the coexistence and integration of diverse life goals among adolescents.

We were also interested in grade and gender differences in the youth’s accounts of life goals. Researchers have suggested that youth explore their identities in more depth in middle-to-late adolescence to address their stronger need for individual uniqueness and autonomy [40]. That is, adolescents in higher grades will identify more with self-oriented life goals, and they will see the acquisition of personal achievement and the expansion of their interests as having an increasingly important meaning in their lives. We thus explored grade differences in youth’s endorsement of life goals in this study, which might contribute to a further understanding of this understudied issue.

Because of gender-stereotypical ideologies, boys in China are expected to be independence- and uniqueness-focused, whereas girls are perceived to be more interested and invested in social relationships and interdependence [18]. Therefore, it was expected that boys would endorse more life goals that are relevant to autonomy and uniqueness in school, family, and other contexts, while girls would perceive values that contribute to social relationships to be more important.

## 2. Methods

### 2.1. Participants

Participants in the study consisted of 129 adolescents (65 males) in Shanghai, an urban municipality, and 34 adolescents (17 males) in three rural regions in the provinces of Liaoning, Henan, and Fujian in China. The adolescents were in 8 middle and high schools, with mean ages of 12.84 years (SD = 0.59) in the 7th grade and 16.22 years (SD = 0.87) in the 10th grade in the urban group, and 12.55 years (SD = 0.89) in the 7th grade and 16.93 years (SD = 0.79) in the 10th grade in the rural group, respectively. In the sample, 98.45% and 94.11% in the urban and rural samples were from intact families, and the others were from families with one parent due to parental divorce, death, or other reasons. Among the adolescents, 67.54% and 36.43% in the two groups were only children, and the others had one or more siblings. The structure and organization of the schools are similar in urban and rural regions. For example, class sizes are similar (approximately 40 students). Schools in both regions follow similar guidelines for the arrangement and schedule of each subject. Academic and social extracurricular activities are also organized according to the standards and regulations set by the Ministry of Education.

### 2.2. Procedure

After obtaining youth assent and parental permission, semi-structured interviews were conducted face-to-face in a private room in the school. Each interview lasted approximately 20 min, and all were administered, audio recorded, and transcribed by a group of Psychology graduate students who were research assistants in China. Prior to the interviews, all research assistants were trained by the PIs of the project on how to conduct high-quality interviews (e.g., building rapport, strategies for reflecting, paraphrasing, clarifying, and summarizing). Weekly meetings involving the PIs and the research assistants were held throughout the study period to ensure that the research assistants were following the interviewing and coding instructions rigorously. This study was approved by the institutional review board of the first author’s university. Participants were encouraged to elaborate and provide more details in their responses, yet they were allowed to refuse to answer any question or withdraw at any point during the study. Written assent was obtained from all participating adolescents, and written consent was obtained from their parents through the school. The participation rate was approximately 98.02%.

### 2.3. Interview Protocol

During each interview, participants were asked the following questions that aimed at obtaining the youth’s accounts for their goals and meaning in life [1,7]: “What are your goals in life?”, “What is the meaning of your life?” and “What do you think the source of happiness in life is?”. When the youth responded to the questions, interviewers were instructed to ask two follow-up questions: “Why is this your goal?” and “What will you do to achieve this goal?”. Although interviewers used a structured sequence of questions, they were trained to vary wording, probe, and follow the participant’s lead when appropriate.

### 2.4. Data Analyses

After the interviews, the researchers transcribed all the recordings, which were encoded and analyzed using the thematic analysis method, a widely used qualitative analytic method in psychology [39]. This method is inductive and theoretically flexible in identifying broad themes that emerge from qualitative data. The PIs and two research assistants first read the transcripts line-by-line to get familiar with the content. Then, 1004 initial fragments of text were identified after a series of discussions among the research team. Then, the two research assistants each coded ten transcripts, and the team compared the thematic codes from both assistants to resolve any coder disagreement, refine the code definitions, and develop a finalized coding manual. After the consensus process was achieved, all the transcripts were coded using the coding manual, and 13 themes were eventually extracted.

Thematic analysis was used to systematically generate initial codes, search for themes, and review potential themes related to the coded data. For example, during the formation process of the theme of Family well-being, when initially being interviewed, many participants (69.33%) described their family life as the source of happiness, as shown in a statement, such as “I’m happy after I return home from school and eat with my family. I feel that the day is very meaningful”. Descriptive codes were then extracted from the interviewees’ statements to capture their life goals. After that, common elements in the codes were extracted to form subthemes. For example, when discussing family life, some participants talked more about being connected with family members, while others talked more about their responsibilities as a part of the family. Therefore, we concluded with 2 subthemes, including physical and emotional closeness and family obligations, which led to the formation of the summarizing theme of Family well-being. The coding manual of the present study and the percentage of adolescents endorsing each theme are presented in Table 1.

## 3. Results

As shown in Table 1, there were 13 themes of life goals, including Family well-being (e.g., “I think the meaning of life is family happiness”), Academic excellence (e.g., “My biggest goal is to get into Tsinghua University and Peking University”), Personal happiness (e.g., “I think it is good to be happy and to have a good time”), Professional achievement (e.g., “My recent goal is to improve my drawing, I want to draw better”), Friendship (e.g., “I think the most meaningful thing in life is to have good friends who will keep you company and comfort you”), Positive interpersonal relationships (e.g., “I think the most meaningful thing is that the class is harmonious and everyone helps each other”), Life experiences (e.g., “I think the goal of life is to accumulate experiences and to participate in a variety of activities”), Kindness (e.g., “I think the meaning of life is to be able to help others and to be able to help others often”), Material wealth (e.g., “I think the most important goal in life is to be rich and be able to buy a lot of things”), Independence (e.g., “I think the goal of life is to find your values, to live your own life, to be able to make your own choices”), Health (e.g., “My goal in life is to be healthy and that my family is also healthy”), Social contributions (e.g., “My ideal is to be a kindergarten teacher, the teacher is very honorable, to contribute to the flowers of our country”), Being alone (e.g., “My own most meaningful and happy moments are when I am alone, listening to songs, drifting off and thinking about nothing”).

### 3.1. A Closer Examination of the Three Most Endorsed Life Goal Themes

For the three most endorsed life goal themes (i.e., Family well-being, Academic excellence, and Personal happiness), we further analyzed the connotations associated with each theme.

Family Well-being. As the most endorsed life goal theme, it was characterized by close physical and emotional ties among family members and a strong sense of family obligation. Specifically, Family well-being centers on respect, support, and care for one’s family and emphasizes family pleasures and filial piety. As reflected in the codes, the former was defined as seeking moments of simply being together or the experiences of sharing activities and fun with family members, whereas the latter was construed as children’s love, respect, support, and deference to their parents and other elder relatives in the family. Sample quotes on the two subthemes of Family well-being were “It is the most meaningful thing to be with parents”; “A family is happiest when they eat together”.

Academic Excellence. In this study, 66.37% of the adolescents reported “having good grades” and “getting into my dream college” as their ultimate goals. This strong emphasis on academic excellence may be attributed to the adolescents’ adherence to their responsibilities as a learner and compliance with the parents’ academic expectations (e.g., “Students’ duty is to learn…”, “Learning well is the only way to get into a good university, which is a must…”, “We must keep up with our learning goals to get good grades and to make constant progress…”). Moreover, academic excellence was regarded as the key to future success and a moral obligation to return the favor to parents’ support and love. For example, over half of the participants voiced that academic achievement “lays the foundation for everything in the future” and “gives me more opportunities, wider connections, and an easier and happier life”. Moreover, these participants shared that “Good grades and college admission are valued by my parents”, “I want to study hard, get into a good university, and let mom and dad live a good life”, “I want to study hard since it is not easy for my parents to raise me, and I must make some achievements and give them a comfortable life later”, and “Work hard, make progress bit by bit every day and get better grades every time; these are the best reward for my parents”.

Personal Happiness. The third most mentioned life goal theme was personal happiness. This theme was identified in the participants’ statements relating to one’s emotional pleasure and positivity (e.g., “Do things that make yourself happy, such as dancing, listening to music, and this is very, very important to me…”, “Be happy and have a good mood, then you can study harder and play harder…” and “Live a smart, comfortable, and happy life…”). In addition to pleasure and enjoyment, the youth’s accounts of personal happiness also indicated relaxation and freedom, which were closely tied to their academic stress (e.g., “I’m doing a lot of schoolwork every day, but I don’t need to think about anything when eating…”, “I just want a happy and carefree life, without so much stress…”, “I want my life to be simple and carefree, like no homework and I can just go shopping with my friends…”, “Happiness is more important than my grades, there is no end to learning, but happiness is the end…”).

### 3.2. Gender, Grade, and Urban-Rural Differences in Chinese Adolescents’ Life Goals

Chi-square analyses were used to examine gender, grade level, and urban-rural differences in the percentages of adolescents who endorsed each identified a key theme of life goals. As shown in Table 2, there were no significant associations between gender and the percentages of adolescents’ life goal endorsement, suggesting that girls and boys may support all the life goal themes equally, χ^2^_(df=1)_ = 0.02 to 1.36, *p* > 0.05. Also, the results showed significant grade differences in adolescents’ endorsement of themes of Family well-being, Personal happiness, Life experiences, Kindness, Independence, and Being alone. Specifically, compared to high-school students, more middle-school students endorsed Family well-being and Kindness, whereas fewer of them endorsed Personal happiness, Life experiences, Independence, and Being alone.

Similarly, there were urban-rural differences in the adolescents’ endorsement of Family well-being, Personal happiness, Life experiences, and Health. Specifically, more rural students endorsed values of Family well-being, whereas more urban students endorsed life goals of Personal happiness, Life experiences, and Health. This difference was, for example, illustrated by more accounts of physical and emotional closeness with the family among the rural group, such as “Staying with families is the best thing” and “Spending time with family members and having fun with them is important. They love me very much, and I am very dependent on them”. Also, rural youth expressed more concerns about their parents toiling in low-paid farming work or stigmatized, dangerous, demeaning jobs in urban cities as migrant workers. They were expected by their migrant parents to take care of other family members (e.g., grandparents) in the household, as shown in the quotes unique to the rural group, e.g., “I help my grandparents so that they do not need to work too hard. This way, my parents will feel less stressed when being so far away”.

## 4. Discussion

The important role of life goals in shaping and guiding adolescents’ social behaviors has been acknowledged in the literature yet little is known about the specific themes of the life goals in contemporary Chinese adolescents, particularly in the context of social change. By having adolescents discuss their life goals, life meaning, and source of happiness in life, the current qualitative study extended the literature by identifying the key themes of life goals in contemporary Chinese adolescents. There were thirteen themes of life goals identified in the present sample and the three most endorsed themes were Family well-being, Academic excellence, and Personal happiness. In addition, cross-grade and urban-rural comparisons on Chinese adolescents’ endorsement of these life goal themes revealed that more middle schoolers and rural students tend to endorse life goals that emphasize a sense of belonging and group well-being, while more high schoolers and urban students tend to favor life goals that underscore individual independence and uniqueness, respectively.

Overall, the thirteen life goal themes extracted from the adolescents’ accounts largely fall into the two major categories of cultural orientations: self-and group-orientations. Specifically, self-orientations entailed life goal themes: Academic excellence, Personal happiness, Professional achievement, Life experiences, Material wealth, Independence, Health, and Being alone, whereas group orientations included themes relating to, Family well-being, Friendship, Positive interpersonal relationships, Kindness, Social contributions. This is consistent with the argument that self-and group orientations appear to coexist in contemporary Chinese adolescents [17,18], which asserts the notion of the “bicultural self” of the Chinese [41]. On one hand, group-oriented life goal themes (e.g., Family well-being, Friendship, Positive interpersonal relationships that are traditionally endorsed by Confucianism continue to guide how Chinese adolescents evaluate their own and others’ social behaviors. Moreover, favorable feedback one receives when displaying group-oriented behaviors may strengthen adolescents’ endorsement of such life goals. On the other hand, individual achievement, independence, and autonomy that reflect self-orientation are also emphasized in Chinese adolescents nowadays. As argued by Liu et al. [18] and Zeng and Greenfield [36], these goals are likely to render opportunities for individuals to enhance their agency, which may in turn contribute to group functioning and belonging. For example, Academic excellence is highly valued in Chinese society and it represents not only individual success but also filial duty affecting the entire family [31,42], where the former reflects self-oriented life goals and the latter, group-oriented life goals. Taken together, these themes of life goals endorsed by the adolescents in the sample reveal an integration of self-orientation into traditional, group orientation in contemporary China [17].

### 4.1. Grade and Urban-Rural Differences

Despite the coexistence of the thirteen life goal themes in Chinese youth, there are grade and regional differences in their endorsement. More high school students endorsed self-oriented life goal themes (e.g., Personal happiness, Professional achievement, Independence), whereas more middle school students endorsed group-oriented life goal themes (e.g., Family well-being, Kindness, Positive interpersonal relationships). This suggests that as youth grow older, there is a stronger personal need to seek independence, autonomy, and individual uniqueness, which is attributed to one major task for adolescents that focuses on identity formation and exploration [43]. Relatively, younger youth are cognitively less capable of navigating their selves and thus need more guidance and support from their socialization agents, including parents, teachers, and other significant adults. In addition, our results seem to echo previous yet limited findings regarding an increase in the importance of having independent thoughts and actions and of seeking fulfillment through personal growth in adolescence [44].

Moreover, more urban youth endorsed self-oriented life goals, whereas more rural youth endorsed group-oriented life goals, suggesting evident urban-rural differences in the Chinese adolescents’ endorsement of specific goals in various areas of life. Due to the comprehensive economic reform, the urban Chinese context is transitioning from a traditional group-oriented value system toward a Western self-oriented value system [18,25]. As parents, teachers, and other socialization agents prepare adolescents for success in the competitive, urban environment, their encouragement of self-assertiveness, initiative-taking, and other self-oriented behavioral qualities is likely to reinforce the urban youth’s belief in expanding their life experiences and pursuing pleasure and happiness [24]. On the contrary, in rural regions where these self-oriented life goals are less present and group-oriented life goals are yet more predominant, youth are expected to exhibit traditional behavioral qualities, such as constraining their behaviors to promote group functioning and prioritizing family well-being over personal pleasure [17]. Many rural adolescents shared that it was their obligation to engage in family assistance behaviors to support their families, such as caring for siblings and completing household chores. This internalization of a strong sense of family obligation among rural adolescents appears to provide them with a means to maintain family closeness and promote family well-being. Overall, our results are in line with the qualitative findings by Liu et al. [18] on the different levels of self and group orientations between urban and rural Chinese children. Yet, the present study provided a closer look at urban-rural differences in the adolescents’ endorsement of specific life goal themes within each orientation.

We found in this study that girls and boys endorsed self-and group-oriented life goal themes equally, which is inconsistent with previous findings that Chinese girls were more group-oriented than boys [18,27]. Perhaps the integration of self-orientation into the traditional group orientation has shifted society’s views on gender roles to be more egalitarian. This argument is supported by Greenfield’s theoretical perspective [34] in linking sociodemographic changes, dynamic cultural values, and gender role socialization. As Manago et al. [45] argued, as a society becomes more urban, commercial, and technologically advanced with more opportunities for formal education, gender roles are idealized as more equal rather than complementary [45]. Females and males are thus expected to receive equal treatment and opportunities to achieve their life goals. More research on boys’ and girls’ priorities of life goals in contemporary China is needed in the future.

### 4.2. Limitations and Future Directions

There were several noticeable limitations in the present study. First, fewer rural adolescents participated in the study than urban adolescents, despite the invitation to participate being open to both groups. In addition, rural adolescents from western mountainous, and other more remote areas of China were not included, which may have exacerbated the rural-urban differences. Similarly, the urban group was from one of the most developed metropolitans in China. It is possible that compared to their rural peers, urban children identify more strongly with life goals (e.g., life experience) because they have more resources and opportunities to develop their personal interests and broaden their life perspectives and experiences. Future research should be conducted using a larger sample size and from a broader range of rural and urban communities.

Second, quantitative research has indicated complicated implications of family-obligation life goals for youth’s academic and psychological well-being [46]. In other words, higher levels of family-obligation life goals may contribute to the adolescents’ experiences of two conflicting sets of behavioral expectations: the increase in family assistance demands versus the increase in school demands [47]. Future studies using a qualitative approach are warranted to explore the rural Chinese adolescents’ responses when facing the home-school conflict or other sets of conflicting life goals. Third, the occurrence of major life events (e.g., acute changes in parents’ physical and psychological well-being, middle to high school transition) is likely to shape the adolescents’ life goals [48]. It would be interesting to investigate life goal stability and change in Chinese adolescents, particularly when they experience changes in life circumstances.

## 5. Conclusions

By using a mixed-methods approach, the present study identified thirteen themes of life goals among adolescents in contemporary urban and rural China, and the most mentioned themes among all were Family well-being, Academic excellence, and Personal happiness. In addition, the results revealed grade and urban-rural differences in the adolescents’ endorsement of the themes. These findings confirm and extend the theoretical categorization of self-and group-oriented life goals in contemporary China by providing specific themes of life goals in each category. This study informs school-based curriculums and programs that aim to help students with life goal-setting about the different themes that could be strengthened for urban and rural youth in order to better align with the youth’s value identity in contemporary China [17].

## Figures and Tables

**Table 1 behavsci-13-00326-t001:** Coding Manual and Percentage of Endorsement.

Theme	Category	Sample Codes	Percentage (%) of Endorsement
Family well-being	Physical and emotional closeness	Spending time with family members Communicating with family members Family vacations Family gatherings	69.33
	Family obligations	Take care of parents and relatives Doing chores to help parents	
Academic excellence	Learning responsibilities	My duty to study hard It is my responsibility to get good school grades	66.37
	Good grades	Good scores on school subjects Improve my academic ranking in the classroom	
	Dream college	Best score on the college entrance exams Studying hard to get into my dream school	
Personal happiness	Emotional pleasure and positivity	Doing things that make me happy Enjoy myself Live a happy life	47.85
	Relief from academic stress	No homework Worry free	
Professional achievement	Sense of fulfillment	Realization of ideals Successful in setting and achieving goals	47.22
	Improving abilities	Improving my learning skills Developing soft skills	
Friendship		The companionship of friends Having sincere friends	38.65
Positive interpersonal relationships	Group atmosphere	Friends in my peer group enjoy each others’ company No conflicts in the classroom	26.99
	Interdependence	Mutual help Mutual acceptance	
Life experiences	First-hand experiences	Traveling around the world Meeting different people	22.70
	Second-hand experiences	Reading books to widen my horizon Learning life lessons from my parents	
Kindness		Be considerate Volunteering to help others	19.63
Material wealth		Making good earnings to get the stuff that I want Having enough money to afford my own space	15.34
Independence	Psychological autonomy	Feeling self-reliant Being able to chase my dreams despite my parents’ disapproval	11.66
	Behavioral autonomy	Doing things independently Making decisions on my own and following through on them	
Health	Good physical health	Playing Basketball regularly to be healthy Having a healthy, balanced diet	11.04
	Mental well-being	Being active in sports to decrease stress Doing sports makes me feel confident	
Social contributions	Benefiting the society	Contributing to my civic society Using my profession to help others after I graduate	10.43
Being alone	Sense of being alone	Feeling comfortable in my own company Enjoying alone time	7.98
	Seeking solitude	Traveling alone to get refreshed Spending time alone to think creatively	

For the quantitative analysis, the participant genders were coded as 1 = boys and 2 = girls, grades were coded as 1 = middle schoolers, 2 = high schoolers, and the residential regions were coded as 1 = urban, 2 = rural.

**Table 2 behavsci-13-00326-t002:** Grade and Urban-Rural Differences in Adolescents’ Endorsement of Each Theme.

Theme	Middle-School (%)	High-School (%)	χ^2^ (df = 1)	Urban (%)	Rural (%)	χ^2^ (df = 1)
Family well-being	81.93	56.25	12.63 ***	63.57	91.18	9.65 **
Academic excellence	66.27	66.25	0.00	64.34	73.53	1.10
Personal happiness	33.73	62.50	13.51 ***	56.59	14.71	18.92 ***
Professional achievement	40.96	53.75	2.67	45.73	52.94	0.56
Friendship	37.35	40.00	0.12	37.98	41.18	0.11
Positive interpersonal relationships	26.50	27.50	0.02	26.36	29.41	0.13
Life experiences	13.35	32.50	8.60 *	26.35	8.82	4.71 *
Kindness	26.51	12.50	5.07 *	18.60	23.53	0.41
Material wealth	13.25	17.50	0.57	16.28	11.76	0.42
Independence	3.61	20.00	10.62 **	11.63	11.76	0.00
Health	14.46	7.50	2.00	13.95	0.00	5.33 *
Social contributions	9.64	11.25	0.11	10.08	11.76	0.08
Being alone	3.61	12.50	4.38 *	8.53	5.88	0.26

Note: * = *p* < 0.05, ** = *p* < 0.01, *** = *p* < 0.001.

## Data Availability

The data presented in this study are available on request from the corresponding author.

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
