# Peer review of "Characteristics of Adolescent Life Goals in Contemporary China: A Mixed-Methods Study"

_behavsci, 2023, doi:10.3390/bs13040326_

Round 1

Reviewer 1 Report

The structure of the article is misleading. The section in bold is called "Introduction". The following sections are in italics: Values of Chinese Adolescents, Life Values of Chinese Adolescents in the Context of Social Change, and Overview of the Present Study. It remains unclear: are they part of the introduction? I recommend that authors reconsider the structure and make it clear where the reader sees the introduction, the contextual background of the study, and its theoretical framework.

The conclusion should be more grounded: why this research is important, what theoretical implications it brought.

The authors should clarify what they mean: "This study represented the first qualitative attempt to explore the implications of social change for adolescents’ life values in Chinese context." Their study is quantitative in nature.

Author Response

Reviewer #1:
1.1 The structure of the article is misleading. The section in bold is called "Introduction". The following sections are in italics: Values of Chinese Adolescents, Life Values of Chinese Adolescents in the Context of Social Change, and Overview of the Present Study. It remains unclear: are they part of the introduction? I recommend that authors reconsider the structure and make it clear where the reader sees the introduction, the contextual background of the study, and its theoretical framework.

Response: Thank you for pointing this out. We restructured the Introduction to more clearly present our theoretical framework and the extant literature on youth values, particularly in contemporary China. We are glad to continue thinking through how to make this clearer if needed.

1.2 The conclusion should be more grounded: why this research is important, what theoretical implications it brought.
The authors should clarify what they mean: "This study represented the first qualitative attempt to explore the implications of social change for adolescents’ life values in Chinese context." Their study is quantitative in nature.

Response: We have further clarified our study approach and implications of the present study on p. 17.

Reviewer 2 Report

Minor grammar/ English language flow issues throughout e.g., line 33 'despite generally stable over time' should read "despite being generally stable over time"; line 111 'the young generations' should read "the younger generation"; and the wording of line 42 'one major developmental task for identity formation'

You need to review the overview of the present study as it is very unclear e.g., in the first paragraph the first sentence: on line 151 to 154 is too long making it hard to follow; the second is also hard to follow; you introduce reasons why you wish to look at gender differences when this should be in the introduction.

The authors need to go back to the seminal source when discussing self-oriented and group-oriented values - line 52-54; the need a page number for the direct quotes in line 113-114. You also rely very heavily on a previous paper by two of the authors rather than the broader values research literature,

Specifically, my main concern is that the authors have clearly mis-quoted/ misinterpreted Schwartz (2012), who they begin their introduction quoting, as they are not investigated the 10 basic values from theory that appear in Schwartz paper. They also seem to have taken 'quotes' from other values researcher that are discussing Schwartz theoretical values and used them to support their reasoning when they are not using the same values theory (e.g., Vecchione et al, Cieciuch et al, and Doring). This leads me to question whether they are also misquoting/ misunderstanding other research in their paper.

The authors discuss 'life values' which they do not define well and then go on to decide the 'life values' based on their 'themes' (which are commonly used in the literature as wellbeing themes e.g., goals, the meaning of life, and happiness) rather than the values in the well accepted values theory of Shalom Schwartz (1992) (which have been tested in more than 80 countries in 100s of studies). Therefore, I cannot endorse this as 'values' research.

I suggest that the 'life values' in the title and the misinterpretations of Schwartz work be removed or the authors reexamine their 'themes' in relation to the values theory that they quote at the beginning of their introduction as well as in the discussion.

Author Response

Reviewer #2:

We appreciated your valuable suggestions!

2.1 Minor grammar/ English language flow issues throughout e.g., line 33 'despite generally stable over time' should read "despite being generally stable over time"; line 111 'the young generations' should read "the younger generation"; and the wording of line 42 'one major developmental task for identity formation'

Response: Thank you for pointing out the errors. We have made corrections and edits to enhance the flow and would like another opportunity to further improve the manuscript if needed.

2.2 You need to review the overview of the present study as it is very unclear e.g., in the first paragraph the first sentence: on line 151 to 154 is too long making it hard to follow; the second is also hard to follow; you introduce reasons why you wish to look at gender differences when this should be in the introduction.

Response: Thank you for your suggestion. We have edited the overview section to clarify and moved the discussion up to the Introduction.

2.3 The authors need to go back to the seminal source when discussing self-oriented and group-oriented values - line 52-54; the need a page number for the direct quotes in line 113-114.

Response: Thank you for your suggestion. We have made changes accordingly on p. 3-4 and 4, respectively.

2.4 You also rely very heavily on a previous paper by two of the authors rather than the broader values research literature, Specifically, my main concern is that the authors have clearly mis-quoted/ misinterpreted Schwartz (2012), who they begin their introduction quoting, as they are not investigated the 10 basic values from theory that appear in Schwartz paper. They also seem to have taken 'quotes' from other values researcher that are discussing Schwartz theoretical values and used them to support their reasoning when they are not using the same values theory (e.g., Vecchione et al, Cieciuch et al, and Doring). This leads me to question whether they are also misquoting/ misunderstanding other research in their paper.

The authors discuss 'life values' which they do not define well and then go on to decide the 'life values' based on their 'themes' (which are commonly used in the literature as wellbeing themes e.g., goals, the meaning of life, and happiness) rather than the values in the well accepted values theory of Shalom Schwartz (1992) (which have been tested in more than 80 countries in 100s of studies). Therefore, I cannot endorse this as 'values' research.

I suggest that the 'life values' in the title and the misinterpretations of Schwartz work be removed or the authors reexamine their 'themes' in relation to the values theory that they quote at the beginning of their introduction as well as in the discussion.

Response: Thank you for pointing this out and our apologies for the misleading in-text citations. We corrected it after revisiting the cited publications and used “values” throughout the manuscript. We have elaborated on Schwartz’s theory of basic values, added a brief discussion on Rokeach’s theory of human values, and discussed how both theories laid the foundation for our theoretical approach of classifying values of Chinese youth into self-and other orientations. We are glad to continue thinking through how to better address it if needed.

Reviewer 3 Report

This is a valuable paper that clearly shows cultural shifts in Chinese values.  There are some suggestions for improvement:

1)  In line 36, "dramatic societal changes" in China are mentioned.  It would help to give a few examples of these changes.

2)  In lines 63-66, when it is explained about how expression of personal desires is incompatible with Confucianist values, there needs to be a citation supporting this information.

3)  In the second paragraph under Values of Chinese Adolescents, it ends with a discussion of group-oriented values but does not discuss individualistic values.  There should be a discussion of those values to complement the discussion of group-oriented values.

4)  Somewhere in this section, it would help to mention the difference between individualistic cultures (i.e. United States, Western Europe) and collectivistic cultures (Mexico, China) traditionally to help frame the difference in values.

5)  In the Overview of the Present Study, it is first introduced that gender and grade level will be studied.  One fact is given about each in the second paragraph.  These variables should be introduced in at least a paragraph in the literature review with more citations given.

6)  Under Participants, there is a large difference in sample sizes between urban and rural adolescents.  This fact should be mentioned as a limitaiton.

7)  Many more urban adolescents were only children, which could affect their push for independence as they did not have siblings to relate to.  This fact also should be mentioned as a limitation.

8)  Under data analyses, traditional thematic analyses starts with codes that themes are built upon.  It looks like from the authors' table that this was done, but it is unclear in the explanation about the process.  This needs to be revised for clarity.

9)  The Data Analyses section also needs to address the quantitative analyses and describe how variables such as gender and urban vs. rural were coded.

10) In the Results section, qualitative results that briefly present all 13 themes with quotations supporting them should come first.  The authors' expanded explanations later on in the paper about the three most common themes could be integrated into this section with the stress on the three most common themes.  It is unclear what the themes mean in the quantitative analyses when the qualitative results are not presented first.

11) In the section that more closely examines the three most common themes, in the part on family well-being, are all of the first group of quotations from urban adolescents?  In line 280 it says, "Some of the rural interviewees....".  This should be clarified.

12) The quotation in lines 282 and 283 does not fit with its previous description.  It discusses being away from home while the previous description discusses caring for relatives at home.

13) In paragraph two of the Discussion, where the themes are divided into types of values in lines 326-330, all themes should be listed to show the reader which values they fit into instead of some themes.

14) The discussion from lines 333-339 spoke of all Chinese adolescents.  Is this true, or do these statements vary by gender and urbanicity vs. rurality as shown in this study?

Author Response

Reviewer #3:
Thank you for your helpful comments!

3.1   In line 36, "dramatic societal changes" in China are mentioned.  It would help to give a few examples of these changes.

Response: We have added a few examples to showcase the dramatic societal changes in contemporary China (p. 2).

3.2   In lines 63-66, when it is explained about how expression of personal desires is incompatible with Confucianist values, there needs to be a citation supporting this information.

Response: We have added a citation to support this argument (p. 3)

3.3  In the second paragraph under Values of Chinese Adolescents, it ends with a discussion of group-oriented values but does not discuss individualistic values.  There should be a discussion of those values to complement the discussion of group-oriented values.

Response: We understand the concern but respectfully disagree. In this paragraph, we focused on group-oriented cultural values and virtues that are central to traditional Chinese society's repertoire of beliefs. In the next paragraph, we discussed the rise of self-oriented values due to the massive economic and social change in contemporary China. 

3.4   Somewhere in this section, it would help to mention the difference between individualistic cultures (i.e. United States, Western Europe) and collectivistic cultures (Mexico, China) traditionally to help frame the difference in values.

Response: Thank you for your suggestion. I have made changes accordingly on p. 2.

3.5 In the Overview of the Present Study, it is first introduced that gender and grade level will be studied.  One fact is given about each in the second paragraph.  These variables should be introduced in at least a paragraph in the literature review with more citations given.

Response: We agree and thank you for this comment. We have added more discussion about the literature on gender and grade/age differences (p. 2)

3.6 Under Participants, there is a large difference in sample sizes between urban and rural adolescents.  This fact should be mentioned as a limitation.
3.7 Many more urban adolescents were only children, which could affect their push for independence as they did not have siblings to relate to.  This fact also should be mentioned as a limitation.

Response: Thank you for your suggestion. We added more discussion about this limitation on p. 16. 

3.8 Under data analyses, traditional thematic analyses starts with codes that themes are built upon.  It looks like from the authors' table that this was done, but it is unclear in the explanation about the process.  This needs to be revised for clarity.

Response: We are grateful for this valuable comment. We have elaborated on Data Analyses to explain the process (p. 6) . We are glad to continue thinking through how to further clarify it if needed.

3.9 The Data Analyses section also needs to address the quantitative analyses and describe how variables such as gender and urban vs. rural were coded.

Response: We have added such information on p. 6.

3.10 In the Results section, qualitative results that briefly present all 13 themes with quotations supporting them should come first.  

The authors' expanded explanations later on in the paper about the three most common themes could be integrated into this section with the stress on the three most common themes.  It is unclear what the themes mean in the quantitative analyses when the qualitative results are not presented first.

Response: We have restructured the Results according to your helpful suggestion p. 7.
3.11 In the section that more closely examines the three most common themes, in the part on family well-being, are all of the first group of quotations from urban adolescents?  In line 280 it says, "Some of the rural interviewees....".  This should be clarified.

Response: We agree with your comment and have clarified the source of the quotations (p. 7). 

3.12 The quotation in lines 282 and 283 does not fit with its previous description.  It discusses being away from home while the previous description discusses caring for relatives at home.

Response: We have added more information to clarify on p. 7.

3.13 In paragraph two of the Discussion, where the themes are divided into types of values in lines 326-330, all themes should be listed to show the reader which values they fit into instead of some themes.

Response: We have listed the specific themes corresponding to the two categories of cultural orientations (p. 15).

3.14 The discussion from lines 333-339 spoke of all Chinese adolescents.  Is this true, or do these statements vary by gender and urbanicity vs. rurality as shown in this study?

Response: The present findings are largely aligned with prior research conducted in different samples of urban and rural youth in contemporary China. It is our contention that the present, mixed-method study provided a closer look at urban-rural and gender differences in Chinese adolescents’ endorsement of the specific value themes.

Round 2

Reviewer 1 Report

The article should end with a clear statement of what their valuable contributions to the field are. A revised version may be accepted.

Author Response

Reviewer #1:
The article should end with a clear statement of what their valuable contributions to the field are. A revised version may be accepted.

Response: Thank you for pointing this out. We make a clear statement of valuable contributions of our study to theory and practices in ‘Conclusion’, p17.

Reviewer 3 Report

The authors answered all reviewer comments adequately.  The paper can be published.

Author Response

Reviewer #3:
The authors answered all reviewer comments adequately.  The paper can be published.

Response: Thank you for your helpful comments.